# Investigation of the Contact Characteristics of Silicon–Gold in an Anodic Bonding Structure

**DOI:** 10.3390/mi13020264

**Published:** 2022-02-06

**Authors:** Lin Zhang, Kaicong Cao, Longqi Ran, Huijun Yu, Wu Zhou

**Affiliations:** 1School of Mechanical and Electrical Engineering, University of Electronic Science and Technology of China, Chengdu 611731, China; castalzhang@126.com (L.Z.); ckcuestc@163.com (K.C.); slash_long7@163.com (L.R.); yuhjuestc@126.com (H.Y.); 2Nuclear Power Institute of China, Chengdu 610213, China

**Keywords:** MEMS, anodic bonding, contact characteristics, contact resistance

## Abstract

Anodic bonding is broadly utilized to realize the structure support and electrical connection in the process of fabrication and packaging of MEMS devices, and the mechanical and electrical characteristics of the bonded interface of structure exhibit a significant impact on the stability and reliability of devices. For the anodic bonding structure, including the gold electrode of micro accelerometers, the elastic/plastic contact model of a gold–silicon rough surface is established based on Hertz contact theory to gain the contact area and force of Gauss surface bonding. The trans-scale finite element model of a silicon–gold glass structure is built in Workbench through the reconstruction of Gauss surface net by the reverse engineering technique. The translation load is added to mimic the process of contact to acquire the contact behaviors through the coupling of mechanical and electrical fields, and then the change law of contact resistance is obtained. Finally, the measurement shows a good agreement between the experimental results, theoretical analysis and simulation, which indicates there is almost no change of resistance when the surface gap is less than 20 nm and the resistance is less than 5Ω, while the resistance changes rapidly after the gap exceeds 20 nm.

## 1. Introduction

Anodic bonding is a silicon–glass electrostatically bonding technology proposed by Romerantz and Wallis, and it can steadily connect silicon wafer or metal pad to glass substrate under applied external heat and electrical energy without the need of adhesives [1,2]. This bonding process, therefore, is widely utilized in MEMS (micro-electrical-mechanical systems) device assembly and packaging due to its low bonding temperature, solid bonding interface and long-term stability [3]. Apart from the mechanical connection between die and substrate, the function of anodic bonding introduces a conductive pad to realize the electrical connection [4,5]. The bonding quality of high-end MEMS devices, therefore, is necessary to be investigated carefully and comprehensively, including the observation of bonding surface [6], interfacial analysis [7,8] and surface morphology [9], among which the contact characteristics exhibit a significant impact on the impedance and stability of electrical signal between the silicon and metal pad. Wang theoretically studied the relationship between resistivity of a silicon–gold connection and doping concentration, and showed that the resistivity was (0.0025 ± 0.0005) Ω·cm [10]. Li constructed a series of metal-semiconductor connect models and measurement methods, including a rectangular transmission line model and circle transmission line model [11]. Jia focused on the mechanical field, thermal field and electrical field coupling contact behavior based on Gauss surface model [12]. Zhu established a micro/nano scale contact point plastic model and experiments and found the contact resistance decreased with the contact gap [13]. Pennec et al. indicated the dependence of contact characteristics on the surface roughness [14]. Ardito et al. concentrated mainly on the surface contact phenomenon of MEMS devices and finished the corresponding physical model under different environmental conditions [15]. Cui et al. established a multi-scale model of rough surface contact and calculated the deformation level of the surface under pressure [16]. Zhang et al. investigated the relationship between contact resistance and geometrical parameters and material constant [17]. Lumbantobing et al. carried out research on the contact surface quality of MEMS accelerometers [18]. Jackson et al. established a rough surface contact elastic model [19]. Kogut et al. studied polysilicon-oxidation-polysilicon contact resistance of MEMS devices [20]. Rezvanian et al. established a three-dimensional contact model of RF MEMS switches [21]. Kim et al. gave a perspective on non-Gauss surface contact model [22].

The above mentioned research investigated the contact characteristics of different MEMS structures from many perspectives. The diversity and non-standardization of MEMS, however, made those results too specific to be applied to other devices or structures; additionally, the silicon-metal contact model involves a scale-span problem from the macro characterization to micro morphology, which was not considered in the current literature. Therefore, a detailed modelling process and analysis is proposed in this paper to form a systematical study on the contact characteristics of a silicon–gold surface in the anodic bonding structure of micro accelerometers. The contact area extraction, modelling method of rough surface, contact resistance calculation and experimental verification are included to evaluate the bonding quality of micro accelerometers.

## 2. Contact Characteristics Analysis

### 2.1. Bonding Structure

The bonding structure this paper studies was extracted from a SOG (silicon-on-glass) micro accelerometer whose SEM is shown in Figure 1. The accelerometer consists of three layers. The top layer is a deeply boron doping silicon structure composed of movable proof mass, comb fingers, folded beams and anchors. The bottom is a glass layer for support. The middle layer is a thin gold (Au) electrode, which is sputtered on glass for electrical connection [23]. The segments marked by numbers are a ‘sandwich’ structure where the gold electrode is used for electrical connection. Figure 2 shows an enlarged one of number 5 segment. The Au is sputtered locally on the surface of glass before the silicon and glass are bonded anodically together. The interface between silicon and glass is formed with a chemical reaction under electrical field, heating and pressure to form a group of Si–O covalent bonds, which can fix the sensitive structure firmly on the glass substrate [24]. This bonding technology has been studied intensively and is not considered in this paper. The Au electrodes play a crucial role in the signal transmission of sensor and establish an electrical connection through a physical contact between the silicon and gold layer. This physical contact is of a great importance to the stability of micro accelerometers.

### 2.2. Model of Contacting Area

The physical contact between the silicon and gold is of a micro/nano scale problem, thus the surface roughness of the structure cannot be neglected. The difference of asperity of different position results in the contacting process very complicated. The deformation of the material is varied from one position to another. Current research indicated that the contact performance, including rigidity, thermal resistance, electrical resistance, assembly accuracy and sealing quality, were highly dependent on the surface roughness, stress distribution and load level of contact [25].

According to the Gaussian rough surface theory and G-W contact model, two rough surfaces in contact can be equivalent to a rough one and a smooth rigid one in contact; the conversion process is shown in Figure 3, where *h* represents the average height of rough surface outline. Based on the Hertz contact, the roughness RMS, *σ*, can be expressed as:(1)$$\sigma =\sqrt{\sigma_{1}^{2}+\sigma_{2}^{2}}$$
where *σ*_1_ and *σ*_2_ represent the roughness RMSs of surface 1 and 2, respectively. Thus, the distribution function of surface 3 can be expressed as:(2)$$\varphi (z)=\frac{1}{\sigma \sqrt{2\pi }}\exp(-\frac{z^{2}}{2\sigma^{2}})$$

The equivalent curvature radius and equivalent elastic module of surface 3 can be expressed as:(3)$$\frac{1}{R}=\frac{1}{R_{1}}+\frac{1}{R_{2}}$$
(4)$$\frac{1}{E}=\frac{1-\nu_{1}{}^{2}}{E_{1}}+\frac{1-\nu_{2}{}^{2}}{E_{2}}$$
where *R*_1_ and *R*_2_ are the curvature radiuses of surface 1 and 2, respectively; *E*_1_ and *E*_2_ are the elastic modules, respectively; and *ν*_1_ and *ν*_2_ are the Poisson’s ratios, respectively.

During the contact process, only the part of *z* > *h* in Figure 3 can form a physical contact, thus the probability of surface contact can be expressed as:(5)$$P(z>h)=\int_{h}^{\infty }\varphi \left(z\right)dz$$

Assuming the peak number of rough surfaces is *n*, the number with contact is:(6)$$m=n\int_{h}^{\infty }\varphi \left(z\right)dz$$

Due to the different level of deformation, different peaks exhibit different deformations, including elastic ones or plastic ones or both. As a result, the plasticity index, *ψ*, proposed by Greenwood and Williamson, is introduced to estimate the contact type of peak deformation [26]:(7)$$\psi =\sqrt{\frac{\sigma }{\delta }}=\frac{E}{H}\sqrt{\frac{\sigma }{R}}$$
where *H* is the hardness of material, and *δ* is the normal deformation quantity, which can be expressed based on the above equation as:(8)$$\delta ={\left(\frac{H}{E}\right)}^{2}R$$

Consequently, the physically contacting area can be expressed as:(9)$$A=n\pi R\left(\int_{h}^{h+\delta }\left(z-h\right)\varphi (z)dz+\int_{h+\delta }^{\infty }2(z-h)\varphi (z)dz\right)$$

Correspondingly, the load of contact can be expressed as:(10)$$W=\frac{4}{3}nER^{1/2}\int_{h}^{h+\delta }{\left(z-h\right)}^{3/2}\varphi (z)dz+2n\pi R\sigma_{s}\int_{h+\delta }^{\infty }\left(z-h\right)\varphi (z)dz$$

As for the silicon–gold contact model, the main parameters are listed in Table 1 [15,27], and the thickness of Au layer is 40 nm. Thus, the equivalent curvature radius is 1515 nm and the equivalent elastic module is 56.02 Gpa. Based on the material hardness of 2.07 Gpa, the appearance of plastic deformation occurs after the normal deformation reaches 2.08 nm. Figure 4 shows the relationship between contact load, area and contact gap. The contact area is proportional to the contact load, which indicates the plastic deformation is very small. Additionally, the area is apparently nonlinear to the gap, which shows that more and more rough peaks start to contact with the decrease in contact gap.

### 2.3. Model of Contacting Resistance

The contact resistance highly depends on the contact state, especially the contact area. So, it is necessary to determine the limit of resistance during the contacting process before the design of the bonding process.

According to the above analysis, surface contact consists of a group of point contacts. The current only can pass through the contacted points [28], and the resistance of a single point can be expressed as:(11)$$R_{C}=\frac{1+0.83(\lambda /a)}{1+1.33(\lambda /a)}\frac{\rho }{2a}+\frac{4\rho \lambda }{3\pi a^{2}}$$
where *R_c_* is the contact resistance; *a* is the radius of contacted point; *λ* is the mean free path of an electron; and *ρ* is the resistivity of material.

When a surface contains many contacted points, the total resistance is not only related to the size of each point, but also to the distribution of contacted points. Furthermore, the total resistance decreases with the increase in the distance of points [29]. As a result, it is difficult to calculate the total resistance in practical engineering because it is impossible to measure the physical distance of points.

The limit of resistance, therefore, needs to be discussed. Assuming that all the contacted points gather together to form a big conductive spot, the contact resistance is the upper limit; while the distance between any two points is long enough, the contact is the lower limit. The two limits are
(12)$$R_{U}=f(\frac{\lambda }{a_{eff}})\frac{\rho }{2a_{eff}}+\frac{4\rho \lambda }{3\pi a_{eff}^{2}}$$
(13)$$\frac{1}{R_{L}}=\sum_{i=1}^{n}\frac{1}{R_{Ci}}$$
where *R_U_* and *R_L_* are the upper and lower limits of contact resistance, respectively. *a_eff_* is the equivalent radius of the big conductive spot. *n* is the number of contact peaks. *R_Ci_* is the contact resistance of the *i*th roughness peak.

Substituting the contact area into the limit equations, we can obtain:(14)$$\frac{1}{R_{L}}=n\left[\int_{h}^{h+\delta }\frac{1}{R_{e}}\varphi (z)dz+\int_{h+\delta }^{\infty }\frac{1}{R_{p}}\varphi (z)dz\right]$$
where
(15)$$R_{e}=\frac{1+0.83(\lambda /a_{e})}{1+1.33(\lambda /a_{e})}\frac{\rho }{2a_{e}}+\frac{4\rho \lambda }{3\pi a_{e}^{2}}$$
(16)$$R_{p}=\frac{1+0.83(\lambda /a_{p})}{1+1.33(\lambda /a_{p})}\frac{\rho }{2a_{p}}+\frac{4\rho \lambda }{3\pi a_{p}^{2}}$$

## 3. Contact Modelling and Simulation

To construct the contact model with a Gauss surface is one of the most different tasks in contact characteristics study. In this paper, the process of establishing the rough surface involved two steps. First, an independent Gauss random number sequence with the same dimensions as surface sequence was constructed. Second, based on the surface autocorrelation function *R*(*l*) and correlation length *T*, a number filter was created to transform the random number sequence into a digital random curve with a convolutional operation.

The random surface is represented by a matrix, {*z*(*x*,*y*)},where *x*,*y* = −(*N*−1)/2, −(*N*−1)/2 + 1, …, (*N*−1)/2, and *N* is the multiple of 2, thus the surface consists of *N* × *N* points, and each point can be expressed as:(17)$$z\left(x,y\right)=\sum_{i==-\frac{N-1}{2}}^{\frac{N-1}{2}}\sum_{j=-\frac{N-1}{2}}^{\frac{N-1}{2}}w\left(i,j\right)g(x+i,y+i)$$
where *z* is the height of each point, *i* and *j* represent the coordinates increment, *g* is the Gauss series and *w* is the function of number filter, which can be represented as:(18)$$w\left(i,j\right)=\frac{2\sigma }{\sqrt{\pi T}}\exp\left[-\frac{2\left(i^{2}+j^{2}\right)}{T^{2}}\right]$$

Based on Equation (17), Figure 5a shows a constructed 9μm × 9μm Gauss rough surface, which is of network form and cannot be used to directly establish a finite element model. A reverse engineering, therefore, is utilized to reconstruct the curve after point processing, polygon forming and surface fitting (Figure 5b). The finished model is depicted in Figure 6, in which the bottom surface of the silicon is rough.

At the initial state, *t* = 0 s, a gap exists between the silicon layer and gold electrode. A displacement of *z* = 44 nm is applied to simulate the contact process. Figure 7 shows the different contact levels during the process of contacting. The red zone and yellow zone represent the contacted parts and noncontacted parts, respectively.

After the contact-induced deformation is introduced into the electrical field formed by external voltage, the contact resistance can be extracted by Ohm’s law through the current density. Figure 8a shows the comparison between simulation and theoretical values of resistance, and Figure 8b illustrates the calculated resistance located in the range of limits. The gap size has little impact on the contact resistance when it is smaller than 20 nm, while the resistance is very sensitive to a gap larger than 20 nm. This conclusion is of great significance to the design of a bonding process.

## 4. Experiments and Discussion

The experiment aims to evaluate the bonding quality of micro accelerometers based on the above contact resistance analysis. The measured contact resistance directly reflects the contact states of between silicon and gold, which highly influences the stability of output voltage and sensing capability. The tested microstructure was fabricated from silicon, Pyrex 7740 glass and gold, and the detailed fabrication process can be found authors’ publication [30]. The fabricated and packaged microstructure are depicted in Figure 9a and b, respectively. The anchors of the sensitive part of Figure 9a are numbered by characters in circle. Additionally, each anchor was connected to the package shell through the gold pad, which was anodically bonded to the silicon structure in Figure 9b. The anchor 1 of fixed fingers was connected to shell leg 2, anchor 2, 3, 4, 5 and 6 to leg 3, 7, 6, 4 and 8, respectively. The leg 1 connected to the shielding box of microstructure.

The resistance of micro accelerometers consists of three types: silicon–gold contact resistance, the resistance of the silicon part and packaging-induced resistance. The resistance of silicon and packaging shell resistance are relatively small and stable due to the same fabrication process and batch production. Thus, the contact resistance of silicon and gold can be determined by measuring the electrical impedance between different pads. The digital multimeter with high accuracy is used to measure the resistances of different pads (Figure 10). From Figure 9b, the resistance between leg 4 and 8 can directly reflect the contact level because there is only one mass block existing and the measured resistance includes two contact ones, package shell and mass one. Table 2 shows the test results of 10 dies. It indicates the resistance of the leg is between 4 Ω and 6 Ω, and that of proof mass is 2–4 Ω. The contact resistance, therefore, should be very small, which is shown in Table 1, and all the test values except that of number 3 are smaller than 1 Ω. The third die exhibits a large resistance, 120 Ω, which represents the contact quality between gold layer and silicon substrate is poor. The microscope is utilized to investigate the contact condition from the bottom of die because the glass substrate is transparent. When the contact condition is poor, the gap between glass and silicon will show a colorful interference fringe. Figure 11 illustrates a comparison of good bonding and poor bonding under microscope.

## 5. Conclusions

The contact characteristics of gold and silicon of micro accelerometers was investigated through an analytical method, finite element method and experiment. It is concluded that the normal contact resistance of bonding is less than 1 Ω, which is smaller than that of package legs and proof mass. This result was verified by the observation of the bonding gap under the microscope and can be used to evaluate the contact quality of anodically bonding, which is very difficult to assess by traditional means.

## Figures and Tables

**Figure 1 micromachines-13-00264-f001:**
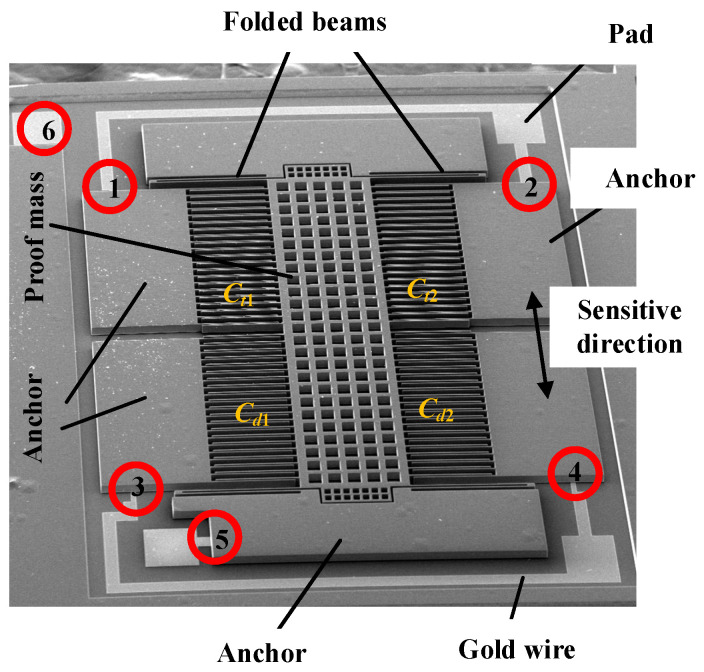
SEM of SOG micro accelerometer.

**Figure 2 micromachines-13-00264-f002:**
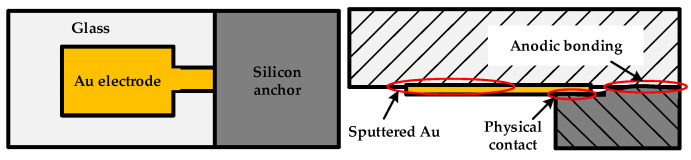
Anodic bonded structure with Au layer.

**Figure 3 micromachines-13-00264-f003:**
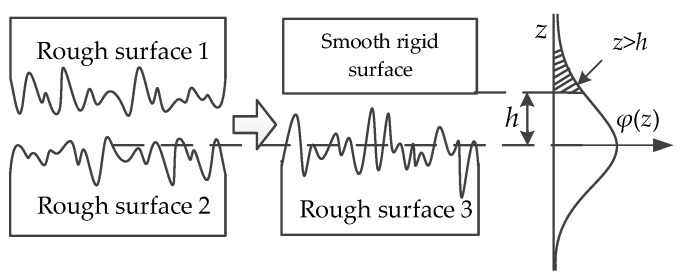
Transferring to a rough surface from two.

**Figure 4 micromachines-13-00264-f004:**
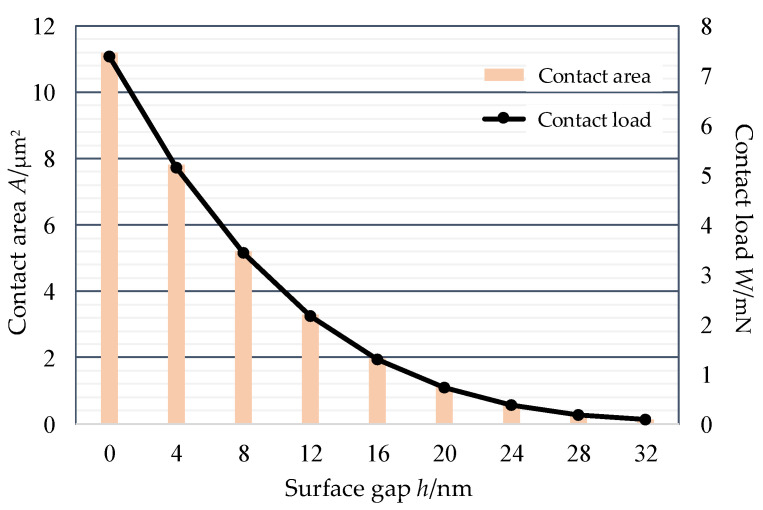
The physical relationship of the contact model.

**Figure 5 micromachines-13-00264-f005:**
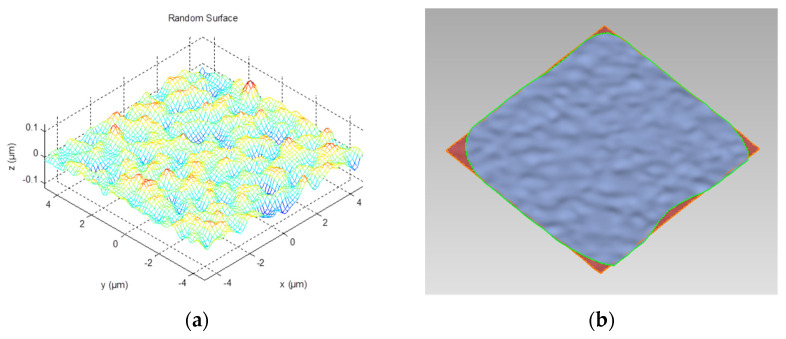
The rough surface model in different stages. (**a**) Gauss rough network. (**b**) Rough surface after fitting.

**Figure 6 micromachines-13-00264-f006:**
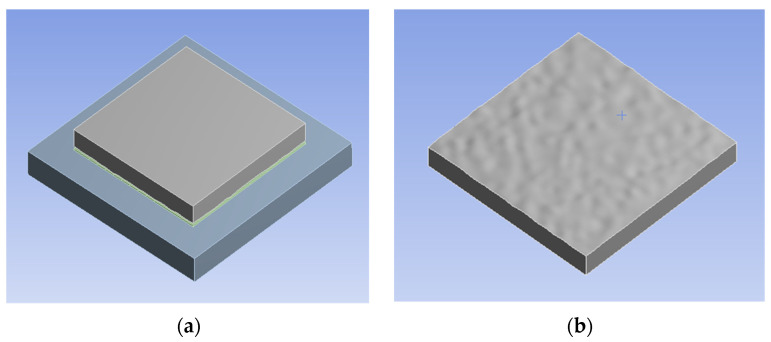
Finite element contact model. (**a**) Contact finite element model. (**b**) Rough surface silicon.

**Figure 7 micromachines-13-00264-f007:**
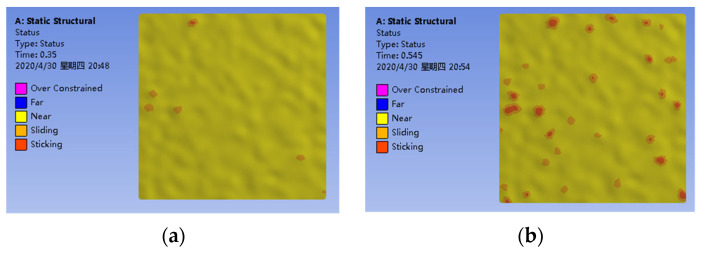
The process of contacting. (**a**) *z* = 16. (**b**) *z* = 24. (**c**) *z* = 32. (**d**) *z* = 42.

**Figure 8 micromachines-13-00264-f008:**
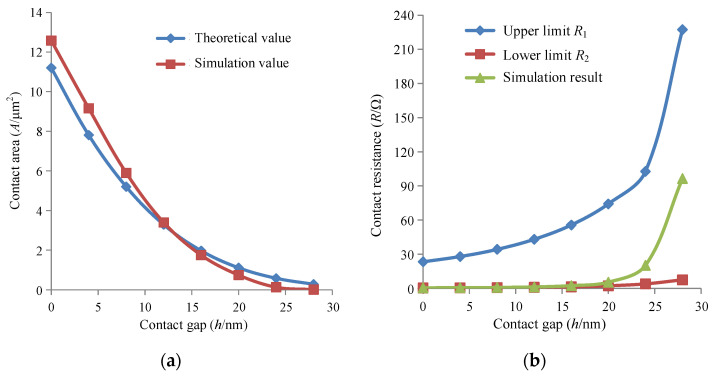
Comparison of the simulation and theoretical value. (**a**) Contact area. (**b**) Contact resistance.

**Figure 9 micromachines-13-00264-f009:**
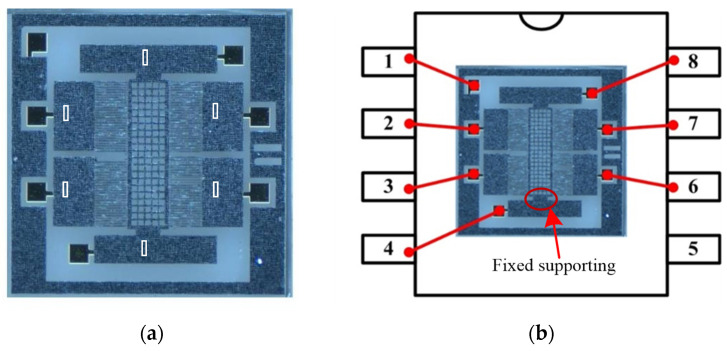
The fabricated and packaged microstructure. (**a**) The microstructure. (**b**) Wire connection of the packaging.

**Figure 10 micromachines-13-00264-f010:**
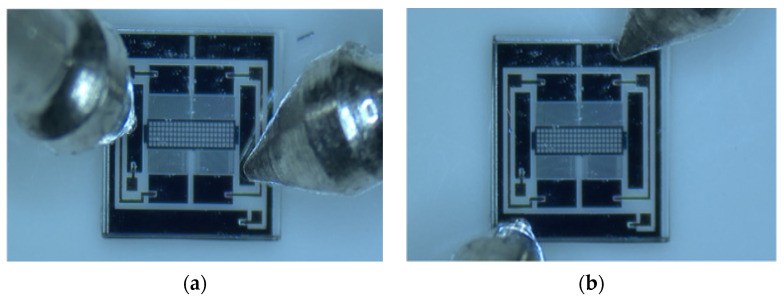
Resistance measurement by multimeter. (**a**) Resistance of mass. (**b**) Resistance of shielding layer.

**Figure 11 micromachines-13-00264-f011:**
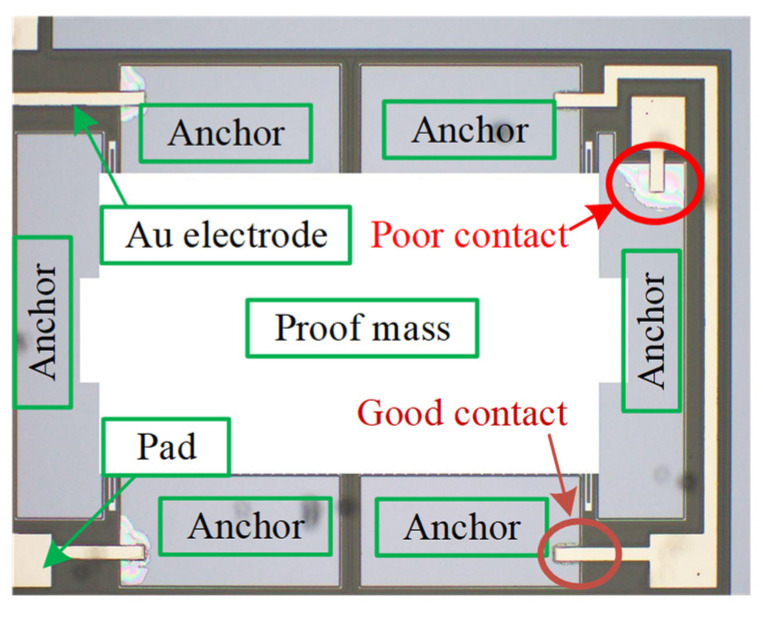
The contact state of bonding under microscope.

**Table 1 micromachines-13-00264-t001:** Main parameters of contact model.

Quantity	*n*	*Σ* (nm)	*E* (GPa)	*R* (nm)	*v*	*σ_s_* (MPa)
Silicon	200	15	169	1515	0.22	/
Gold (Au)	/	0.7	74.46	/	0.3	660

**Table 2 micromachines-13-00264-t002:** Test resistance of 10 dies.

No. #	Leg Resistance (Ω)	Mass Resistance (Ω)	Contact Resistance (Ω)
1	4.68	3.26	0.71
2	5.14	3.12	0.99
3	243	2.98	120.01
4	4.52	3.11	0.71
5	4.66	3.05	0.81
6	4.63	3.38	0.63
7	5.10	3.71	0.70
8	4.89	3.00	0.95
9	4.82	3.63	0.60
10	5.03	3.51	0.76

## Data Availability

Not applicable.

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
