# Peer review of "Investigation of the Contact Characteristics of Silicon–Gold in an Anodic Bonding Structure"

_micromachines, 2022, doi:10.3390/mi13020264_

Round 1

Reviewer 1 Report

The idea to propose a new model to study on the contact characteristics of silicon-gold surface in anodic bonding structure of micro accelerometers is good and it is possible to be somehow useful in the design of the bonding process, however the utility has to be better underlined.

You could identify the defect using the microsope- not by simulations. The simulations did not help you to predict the defect, or tominimize it.

Please clearly explain what is new in your model.

Other aspects to be clarified:

Line 39- the unit for resistivity is Ω·cm2 ?

Line  134-   ℘ is the resistance, not the resistivity ?

Lines 143-145

Why when points gather together to form a big conductive spot, the contact resistance is higher? It should be low (large area --> low R) , and when the distance between the contact points is high it should be high.

Author Response

Dear Professor,

Thank you so much for your time and comments.

Your comments are very useful and constructive. The following content is our response to your advice, and we hope the correction based on your comments will make the manuscript more suitable for publication in “Micormachines”.

Comment: The idea to propose a new model to study on the contact characteristics of silicon-gold surface in anodic bonding structure of micro accelerometers is good and it is possible to be somehow useful in the design of the bonding process, however the utility has to be better underlined.

Response: Thank you so much for kind comments. The following content is our response to your suggestion on the manuscript in details. The corresponding revised content is highlighted in yellow.

Comment: You could identify the defect using the microsope- not by simulations. The simulations did not help you to predict the defect, or to minimize it.

Response: Thank you for your advice. Indeed, the microscope observation is much useful to identify the defect, and actually the experiments have been carried and shown in the last part of manuscript, like Figure 11. May be there is some misunderstanding due to the poor expression. We have revise the total manuscript for a clearer expression. Meanwhile, the simulation is also usefully for defect identification to realize a more comprehensive conclusion, because of the time consuming and high cost of experiments.

Comment: Please clearly explain what is new in your model.

Response: Thank you for your comments. The basic theory of model is no new, which is no easy job. What is new of manuscript includes two aspects. One is the modeling method. We use the bottom-to-top method to construct the model from point to a 3-D model. The other is to apply the model to the gold-glass contact evaluation, which was not investigated in public literature. We think the new application can be recognized as a new aspect for a practical engineering.

Comment: Other aspects to be clarified: Line 39- the unit for resistivity is Ω·cm2?

Response: Thank you so much for your careful review. Sorry for the type, and the square is deleted.

Comment: Line 134- ℘ is the resistance, not the resistivity?

Response: Thank you so much for your careful review. It should be resistivity.

Comment: Lines 143-145. Why when points gather together to form a big conductive spot, the contact resistance is higher? It should be low (large area --> low R), and when the distance between the contact points is high it should be high.

Response: Thank you so much for your careful review. The resistance is related to resistivity, wire length and sectional area. Obviously, for a certain structure, the resistivity, wire length and total sectional area are same, so the total resistance ideally is a constant. However, when the contact points gather together, there exists a coupling between points to form an extra resistance to disturb the flow ability of current, which will result in a higher effective resistance.

Thank you again for your time and consideration.

Best regards.

Sincerely,

Wu Zhou

Reviewer 2 Report

This paper mainly studied the relationship between the resistance and surface roughness under anodic bonding, After simulation analysis and experimental demonstration, there has a relatively low resistance when the surface gap is less than 20nm. I think the novelty of the presented concept and the manuscript quality can meet the publishing requirements in Micromachine, but still have some problems that need to revise.  Here are some comments and I would recommend the acceptance of this manuscript after major revision.

Author Response

Dear Professor,

Thank you so much for your time and comments.

Your comments are very useful and constructive. The following content is our response to your advice, and we hope the correction based on your comments will make the manuscript more suitable for publication in “Micormachines”.

Comment: This paper mainly studied the relationship between the resistance and surface roughness under anodic bonding, after simulation analysis and experimental demonstration, there has a relatively low resistance when the surface gap is less than 20nm. I think the novelty of the presented concept and the manuscript quality can meet the publishing requirements in Micromachine, but still have some problems that need to revise. Here are some comments and I would recommend the acceptance of this manuscript after major revision.

Response: Thank you so much for your kind comments. We will revise carefully manuscript according to your suggestions and comments. The corresponding revised content is highlighted in green.

Comment: 1. In table 1, please provide the reference of the main parameters of the contact model. By the way, the elasticity modulus cannot use here directly because the elastic modulus of the sputtering gold layer is decided by the sputtering process and the elastic modulus of the silicon layer on a silicon wafer also depends on silicon wafer growing modality. If there have no reference, for example, the data of ‘the peak number of surfaces roughness’ and ‘the roughness RMS, σ’, please demonstrate by AFM or relativity test.

Response: Thank you so much for your suggestions. Sorry for the missed citations. We have added them to the revised version. The parameters for silicon and gold were from ‘15. Ardito, R.; Corigliano, A.; Frangi, A. Finite Element modelling of adhesion phenomena in MEMS. Proceedings of 11th Inter-national Thermal, Mechanical & Multi-Physics Simulation, and Experiments in Microelectronics and Microsystems, Bor-deaux, France, 2010; 1-6.’ and ‘27.Yi, T.; Xing, P.; Zheng, F.; Xie, J.; Li, C.; Yang, M. Preparation of Continuous Gold Nano-films by Magnetron Sputtering. Atomic Energy Science and Technology 2010, 44, 479-483.’, respectively.

Comment: 2. How is the Gauss rough network generated in FIG 5a? please provide the details. It is better to describe the details of the generation process.

Response: Thank you for your suggestions. The detailed process has been added to the revised manuscript. The network was generated in Matlab software. First, an independent Gauss random number sequence with the same dimensions as surface sequence is constructed. Second, based on the surface autocorrelation function R(l) and correlation length T, a number filter is created to transform the random number sequence into a digital random curve with a convolutional operation.

Comment: 3. Line 194-196 “The tested microstructure is fabricated from silicon, Pyrex 7740 glass and gold, and the detailed fabrication process can be found authors’ publication [29].” but I can not find the detailed fabrication process in reference 29.

Response: Thank you very much for your careful review. The right publication has been cited to replace current one. The new reference is ‘30. He, J.; Xie, J.; He, X.; Du, L.; Zhou, W. Analytical study and compensation for temperature drifts of a bulk silicon MEMS capacitive accelerometer. Sensor Actuat. A-Phys. 2016, 239, 174–184.’

Comment: 4. The main problem of this paper is that there has a picture with high similarity with previous publications. That is not allowed in scientific publications.

Response: Thank you so much for your careful review. As for the copied pictures, we have reason to argue and beg for your understanding. The object-micro accelerometers this manuscript dealt with is the same as what we studied before. The previous publication (Chen, L.; Li, L.; Zhan, L.; Chen, Q.; He, X.; Yu, H.; Zhou, W. Investigation of Drift Phenomenon in Closed-Loop Capacitive Micro Accelerometers. J. Micromech. Microeng. 2020, 30, 095009.) just showed the tested method and results which are used for state the system drift phenomenon, without any mechanism investigation on the contact resistance. As a result, this manuscript is aiming to find out the behind cause for the tested resistance which is acquired by the same method and equipment of previous publication. The copied pictures have no meanings of novelty and materials but just provide a process of experiments. We hope this is allowed in public literature. We hope you can understand our explanation.

Thank you again for your comments and advice.

Best regards,

Sincerely,

Wu Zhou

Round 2

Reviewer 2 Report

Please try to replace the same picture even though they are focusing on different studies. with this small revision, I think this paper can meet the publishing requirements in Micromachine.